# A Review of the Performance and Benefits of Mass Timber as an Alternative to Concrete and Steel for Improving the Sustainability of Structures

Joseph Abed [1], Scott Rayburg [1,*], John Rodwell [2] and Melissa Neave [3]

1   Department of Civil and Construction Engineering, Swinburne University of Technology, Hawthorn, VIC 3122, Australia; abedjoey@gmail.com
2   Department of Management & Marketing, Swinburne University of Technology, Hawthorn, VIC 3122, Australia; jrodwell@swin.edu.au
3   School of Global, Urban and Social Studies, RMIT University, Melbourne, VIC 3001, Australia; melissa.neave@rmit.edu.au
*   Correspondence: srayburg@swin.edu.au; Tel.: +61-(3)-9214-4944

**Abstract:** The construction industry represents one of the greatest contributors to atmospheric emissions of $CO_2$ and anthropogenic climate change, largely resulting from the production of commonly used building materials such as steel and concrete. It is well understood that the extraction and manufacture of these products generates significant volumes of greenhouse gases and, therefore, this industry represents an important target for reducing emissions. One possibility is to replace emissions-intensive, non-renewable materials with more environmentally friendly alternatives that minimise resource depletion and lower emissions. Although timber has not been widely used in mid- to high-rise buildings since the industrial revolution, recent advances in manufacturing have reintroduced wood as a viable product for larger and more complex structures. One of the main advantages of the resurgence of wood is its environmental performance; however, there is still uncertainty about how mass timber works and its suitability relative to key performance criteria for construction material selection. Consequently, the aim of this study is to help guide decision making in the construction sector by providing a comprehensive review of the research on mass timber. Key performance criteria for mass timber are reviewed, using existing literature, and compared with those for typical concrete construction. The review concludes that mass timber is superior to concrete and steel when taking into consideration all performance factors, and posits that the construction industry should, where appropriate, transition to mass timber as the low-carbon, high performance building material of the future.

**Keywords:** mass timber construction (MTC); cross laminated timber (CLT); engineered timber; green buildings; tall timber buildings; sustainable design; renewable materials

## 1. Introduction

Climate change is one of the most pressing issues facing humanity today [1]. It is well established that greenhouse gas (GHG) emissions caused by human activities are directly linked to the measured increase in the average temperature of the planet. The average global temperature is currently around 1 °C above pre-industrial levels and will continue to climb based on business-as-usual practices [2]. Reports from the Intergovernmental Panel on Climate Change (IPCC) conclude that if measures are not taken to reduce emissions and keep average temperatures below 2 °C above pre-industrial levels, the consequences for Earth's ecosystems will be extremely severe and devastating [1]. These consequences will be exacerbated by the rapid rate of population growth and urbanisation around the world [3]. Current estimates suggest that 68% of the world's population will live in urban areas by 2050 and more than three billion people will need new housing [4]. The prospect of such

vast amounts of new housing is a serious issue because the construction industry already accounts for close to 40% of global $CO_2$ emissions and is a major source of environmental degradation for most countries around the world [5]. Hence, more innovative solutions are required to provide essential building infrastructure in a manner that reduces emissions and helps to mitigate the worst effects of climate change.

The building industry will continue to have a negative impact on the environment if conventional construction practices are maintained [3]. The most recent projections indicate that by 2060 energy demand in the global buildings sector will increase by 30% and $CO_2$ emissions will increase by 10% if efforts are not made to implement low-carbon and energy-efficient solutions for building and construction [6]. In recent years there has been a conscious effort to improve the energy efficiency of buildings and thus reduce emissions related to their operation and maintenance [7]. However, there has been less focus on utilising more sustainable materials to lower the emissions embodied in buildings, which plays a key role in reducing environmental harm [7]. Yet material selection can have a greater influence on a building's lifetime energy emissions (embodied and consumed) than building operation [8]. Therefore, sustainable material selection is an important step towards reducing building related emissions.

The most widely used materials in the modern construction of mid- to high-rise buildings are concrete and steel [7]. These two materials have dominated construction for centuries because of their favourable properties, such as structural adequacy, durability, fire performance and cost [7]. Unfortunately, the production processes for these materials are highly energy intensive and result in significant GHG emissions. For example, it is estimated that for every tonne of cement or steel produced, around 1 tonne and 1.85 tonnes of $CO_2$ are emitted, respectively [9,10]. As such, cement production currently accounts for just over 8% of global $CO_2$ emissions and the steel industry is responsible for around 7–9% of global $CO_2$ emissions [9,10]. Both of these materials are extensively used by the construction industry, with Yan et al. (2010) estimating that 82–87% of total GHG emissions related to building construction are a result of the embodied emissions of the building materials and that 94–95% of these are from the use of concrete and steel [11]. Given the high carbon footprint associated with concrete and steel production, it is critical that alternative materials are used to deliver a net-zero built environment [12].

There continues to be intensive research into ways to reduce the emissions associated with concrete production. One promising outcome of this research within the last few decades has been the development of Geopolymer Concrete (GPC) [13]. GPC replaces traditional Ordinary Portland Cement (OPC) in concrete with industrial by-products such as blast furnace slag and fly-ash. Given that the production of OPC is by far the most emissions intensive part of the concrete production process, the use of recycled materials in GPC has been shown to reduce $CO_2$ emissions by up to 80% [14]. Moreover, GPC exhibits similar or better strength, durability, fire resistance and cost when compared with conventional concrete [13]. However, despite this new method for creating a greener concrete, the manufacture of GPC still requires non-renewable materials that will eventually become unavailable. For example, fly ash (which is a by-product of burning coal) will disappear if we successfully transition to renewable energy and away from coal power. As such, there is a need to find alternatives to concrete and steel for construction, where renewable materials, such as timber, may represent a viable alternative [7].

Mass Timber Construction (MTC) is a term used to describe a family of massive engineered wood products that have structural applications within buildings and that have been proposed as alternatives to concrete and steel within the building industry. One of the many benefits of timber is that it is one of the only renewable building materials that not only reduces emissions but creates negative emissions through carbon sequestration [15]. The use of MTC in building projects has grown in recent years, and there are a number of buildings around the world that showcase this innovative material [7]. However, despite the many advantages of MTC and its potential to reshape the built environment, the building industry continues to use reinforced concrete on a large scale. One of the key

reasons why MTC is yet to be widely adopted in the market is that there is a perceived lack of knowledge and an uncertainty of the risks associated with this methodology [16]. The construction industry is highly risk averse and, as such, new and innovative building materials or disruptive technologies require a significant amount of research to demonstrate that engineering requirements can be met. Hence, to guide decision making and better inform industry professionals, it is crucial that all aspects of MTC are well understood, and the advantages of this modern building material are clearly outlined.

This paper reviews our current understanding of the use of MTC as a building material. The primary goal is to review whether MTC offers a viable alternative to concrete and steel when considering key performance criteria for buildings. As such, this research has the following objectives:

1. Identifying the types of mass timber available;
2. Investigating how mass timber performs relative to the key criteria that drive building material selection, and
3. Comparing the performance of mass timber against the two most commonly used building materials (i.e., concrete and steel).

## 2. Mass Timber Types and Performance

To assess the potential of using mass timber for buildings, a review of existing literature investigating these products was undertaken. This review began with the identification of the most widely used and researched mass timber products currently available within the building industry. Once these had been established, the characteristics of each mass timber product were investigated, including their historical context, manufacturing processes and the advantages/disadvantages of their use relative to key criteria that commonly drive the selection of building materials. These key criteria include structural, environmental, seismic, wind, and fire performance, health benefits, and cost. The expected structural, seismic and fire performance requirements reflect those specified by the International Building Code (IBC). Once acquired, the performance information was used to produce a preliminary assessment of the appropriateness of replacing concrete and steel with engineered timber.

A review of peer-reviewed and industry-related literature reveals that mass timber is a broad term describing a family of massive, engineered wood products that can be used as the primary structural material within buildings [15]. Mass Timber Construction (MTC) refers to a construction process wherein the structural system of the building is predominantly comprised of timber [17]. The main types of mass timber products used in building construction, and those considered in this review, are Cross-Laminated Timber (CLT), Glued Laminated Timber (Glulam), Nail Laminated Timber (NLT), Dowel Laminated Timber (DLT) and Structural Composite Lumber (SCL) (Figure 1). To appreciate the advantages and disadvantages of mass timber more generally, the different types are described and discussed within the context of their performances relative to the key criteria.

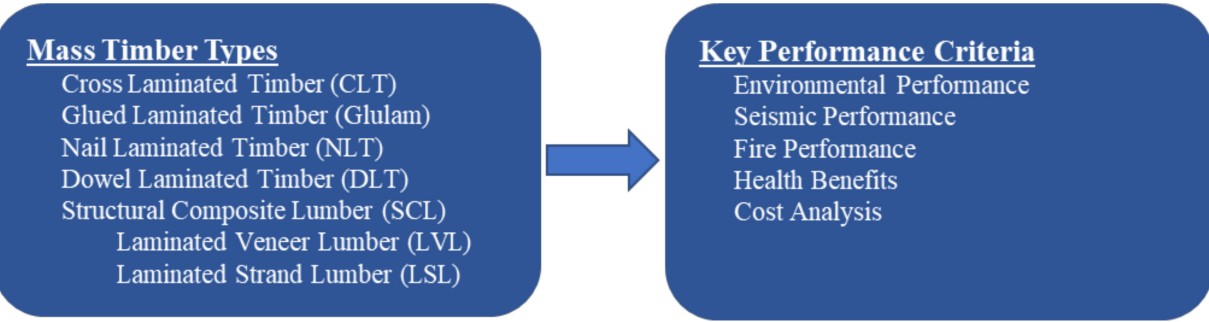

**Figure 1.** A summary of the mass timber types and key performance criteria considered in this study.

### 2.1. Cross Laminated Timber

Cross Laminated Timber (CLT) is a relatively new and innovative mass timber product that is gaining popularity within the construction industry [18]. CLT was developed in Europe in the 1990s and since then extensive research and development into this product has led to its increased use in building projects around the world. The main reason that CLT has garnered so much attention in recent years is that its technical capabilities and environmental properties allow for the use of timber in a wider range of applications than was previously possible.

CLT panels (Figure 2a) are made by stacking layers of lumber boards oriented at right angles to one another and gluing them together with a structural adhesive [18]. The raw timber used in the panels is machine stress rated and kiln dried to a 12% moisture content [15]. Any knots or other defects are removed, and the boards are finger-jointed together to produce the specified lengths. The stack is then placed into a press and face-bonded under pressure. Once removed from the press, the panels are trimmed to exact size and edge profiled using Computer Numerical Control (CNC) machinery and are ready to be delivered to the site. These panels are usually fabricated with an odd number of layers, with three, five and seven layers being the most common. CLT panels vary in size depending on the manufacturer, though they can be made up to 18 m long by 5 m wide with a thickness of up to 500 mm, which makes them ideal for floors, walls, and roofs.

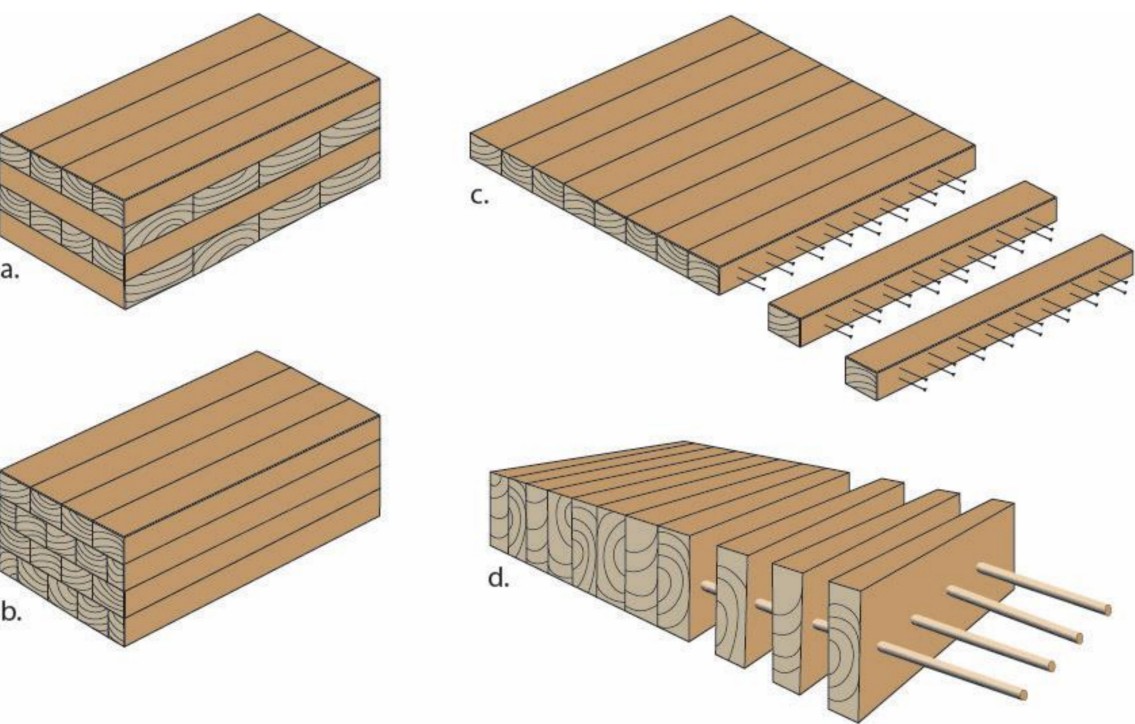

**Figure 2.** The main mass timber types investigated in this study: Cross-Laminated Timber (**a**); Glued Laminated Timber (**b**); Nail Laminated Timber (**c**); Dowel Laminated Timber (**d**).

From an engineering perspective, CLT offers many advantages that make it a viable alternative to concrete and steel for building applications [18]. The method of placing alternating layers of timber crosswise to each other gives the product a high level of dimensional stability, which allows for the prefabrication of large wall and floor elements [19]. Additionally, the cross-lamination process and the large thickness of the panels gives CLT exceptional strength and stiffness as well as two-way span capabilities similar to a reinforced concrete slab. The use of CNC technology allows for high precision and tight dimensional tolerances for each panel, reducing material waste and increasing manufacturing speed.

## 2.2. Glued Laminated Timber

Glued Laminated Timber (Glulam) is a mass timber product that has the potential to be used in a wide variety of applications [20]. Originating in Germany around the 1900s, Glulam was adopted in Australia in the 1950s but is still not as frequently used in this country as it is in Europe and North America. However, this product is gaining traction within the building industry due to its appropriate technical properties and the growing need to improve sustainable practices.

Glulam is composed of multiple layers of dimensional lumber in which the grain of the laminations runs parallel to the length of the member (Figure 2b). The individual pieces of lumber are graded for strength based on their performance characteristics and are bonded together with a durable, moisture-resistant adhesive. Individual laminates are generally finger-jointed to produce greater lengths in accordance with design requirements. Glulam members can vary in size depending on the manufacturer, but are generally 180–630 mm thick, 66–200 mm wide and can be manufactured up to 50 m in length, making them suitable for use as beams and columns. Although variable, the length of the member is usually restricted by transport and handling limitations [20].

One of the key advantages of the Glulam wood product is that it can be manufactured in large sizes and complex shapes that can meet both architectural and structural design requirements. Similar to CLT, Glulam has excellent strength and stiffness properties and a very high strength-to-weight ratio, meaning that by weight it is stronger than structural steel. Given that Glulam is made up of a number of laminates, the strength-reducing properties of each timber element are minimised and the result is a product that is stronger and more reliable than traditional solid lumber [20].

## 2.3. Nail Laminated Timber

Nail Laminated Timber (NLT) is an engineered wood product that was first used in construction over a century ago and is undergoing a resurgence as part of the modern shift towards sustainable materials [21]. Although it was historically used in the construction of warehouses and factories, NLT provides opportunities for a range of modern building applications due to its aesthetic and favourable performance characteristics.

NLT is created by placing individual pieces of dimensional lumber next to one another on edge and fastening the laminations together using nails (Figure 2c) [21]. Using this method of mechanically laminating the dimensional lumber together creates one solid structural element which can be used for floors, roofs, walls and elevator shafts within buildings. The dimensions of the laminations can vary, with standard sizes such as 2-by-4 and 2-by-6 used in timber framing being the most commonly used. Architects have found this type of mass timber product particularly useful given that the monolithic slab nature of NLT offers opportunities to implement unique forms such as curves and cantilevers.

A key advantage of NLT compared with other types of mass timber is that it does not require a dedicated manufacturing facility or any specialised equipment to create [21]. NLT systems can be put together on site using basic carpentry techniques and locally available wood species. Some mass timber suppliers may offer prefabrication of NLT panels for larger and more complex projects; however, it is generally not necessary. Consequently, architects and designers using NLT for recent building projects have found that the product has lower costs and faster procurement times than other mass timber products [22]. Another advantage of NLT is that because it has been used in buildings for over a century, the engineering requirements and details are well supported by building codes and standards. This can provide a significant advantage over newer mass timber products such as CLT, which are not as well understood in building codes and which designers may not be as confident in using.

### 2.4. Dowel Laminated Timber

Dowel Laminated Timber (DLT) is a lesser-known mass timber product that is commonly used in Europe but is slowly gaining traction in North America and other countries [23]. The modern design of DLT was developed in Switzerland in the 1990s as an alternative to mass timber products that utilise metal fasteners, such as NLT. The result is a versatile product that is relatively easy to manufacture while maintaining structural adequacy.

The manufacture of DLT uses a very similar concept to NLT; however, instead of nails or screws, wooden dowels are used to join the timber members [23]. To produce DLT panels, multiple boards of softwood lumber are placed next to one another on edge and are friction-fit together using hardwood dowels (Figure 2d). Once inserted into the timber lamellas, the drier hardwood dowels expand into the surrounding lumber in order to reach moisture equilibrium, which creates a tight friction fit that increases the dimensional stability of the panel. DLT panels are generally manufactured using CNC machinery, which essentially automates the process and produces a highly consistent product in a manner that is safer than conventional manufacturing. The replacement of metal fasteners or adhesives with wooden dowels in the fabrication of DLT panels makes it the only all wood mass timber product, which can offer a number of advantages over other forms of mass timber, such as lower use of adhesives.

The use of adhesives in mass timber products such as CLT and Glulam can have a negative effect on the environment given that they emit toxic gases such as formaldehyde and Volatile Organic Compounds (VOCs) [23]. Hence, the lack of adhesives used in DLT products can create a healthier indoor environment by improving the air quality and reducing the likelihood of allergic reactions. Further, by replacing the adhesives and metal fasteners used in other mass timber products, DLT has the potential to improve the recyclability and reusability of the timber [23]. However, further research is required to better understand the quantitative benefits of DLT with regards to structural and environmental performance.

### 2.5. Structural Composite Lumber

Structural Composite Lumber (SCL) is a term used to describe a family of mass timber products that are characterised by gluing together smaller pieces of wood to create one solid structural member [24]. The two main types of SCL products that are commonly used in building projects are Laminated Veneer Lumber (LVL) and Laminated Strand Lumber (LSL). These products are commonly used in building projects throughout North America with LVL also being widely used in Australia.

LVL was developed in the 1970s and is the most commonly used SCL product [24]. SCL is manufactured by gluing together specially graded, thinly sliced wood veneers under high heat and pressure. Before they are laminated, the veneers are dried, and the associated grains are oriented parallel with the length of the member. LSL is a more recent SCL product that is increasing in use. LSL is very similar to LVL with the major difference being that LSL uses timber strands rather than wood veneers. Both of these smaller SCL products are suited to residential construction and can be used in a range of structural applications including beams, joists, studs and rafters.

The greatest advantage of SCL is that the material is much less prone to dimensional changes compared with conventional sawn timber, meaning that it is unlikely to undergo warping, splitting or shrinking [24]. SCL is also stronger, more reliable and able to withstand greater loads than similar sized conventional timber members. However, this type of mass timber product is generally not well suited to tall buildings and is likely best suited for low-rise construction projects.

### 2.6. Summary

This section provides a comprehensive review of the most commonly used mass timber products, which serve slightly different purposes and have varying benefits depending on their intended use. The most suitable applications for these mass timber products and their advantages and disadvantages relative to one another are provided in Table 1.

**Table 1.** A summary of the relative advantages and disadvantages of using different mass timber products in construction.

| Mass Timber Product | Applications | Advantages | Disadvantages |
|---|---|---|---|
| Cross Laminated Timber (CLT) | Floors, walls, roofs | High dimensional stability<br>High strength and stiffness<br>Easy to manufacture | Higher cost |
| Glued Laminated Timber (Glulam) | Beams, columns | High strength and stiffness<br>Structurally efficient<br>Can be manufactured in complex shapes | Higher cost |
| Nail Laminated Timber (NLT) | Floors, walls, roofs | No specialised equipment required to manufacture<br>Cost effective<br>Fast procurement times | Labour intensive<br>Greater chance of human error |
| Dowel Laminated Timber (DLT) | Floors, walls, roofs | High dimensional stability<br>Easy and safe to manufacture<br>No adhesives or metal fasteners required | Limited panel sizes<br>Limited thicknesses |
| Structural Composite Lumber (SCL) | Beams, columns, joists, studs, rafters | Not prone to shrinking, splitting or warping<br>Able to withstand greater loads than solid timber | Limited panel sizes<br>Limited thicknesses<br>More suitable for low-rise buildings |

## 3. Environmental Performance

A number of factors must be taken into consideration when assessing the environmental consequences of Mass Timber Construction (MTC). Amongst these are greenhouse gas (GHG) emissions, sustainable forestry practices, and end-of-life scenarios for mass timber products. Each of these factors are explored and discussed in the following sections.

### 3.1. Greenhouse Gas Emissions

Timber is one of the few natural, renewable and structural building materials available [15]. Whereas conventional materials, such as concrete and steel, are responsible for vast amounts of $CO_2$ during production, trees naturally remove around two tonnes of $CO_2$ from the atmosphere to create one tonne of their own dry mass [15]. Hence, when mass timber products are used in buildings the carbon sequestered during production is stored over their lifespan. As such, the use of mass timber to replace concrete and steel will drastically reduce the emissions embodied in buildings [25].

In addition to reducing embodied emissions, Mass Timber Construction (MTC) has the potential to reduce emissions related to the construction and operation of buildings [15]. Prefabricated timber construction can greatly reduce emissions produced by heavy vehicles transporting materials, with one study estimating a net saving of 20% compared with traditional methods [26]. Given that mass timber is generally prefabricated offsite, the simple assembly processes and reduced machinery onsite can lead to significant reductions in noise and dust pollution and minimal site waste [15]. Further, mass timber has a high level of airtightness and a low coefficient of thermal conductivity, which improves the energy efficiency of buildings [27]. Another benefit of mass timber is that it lowers strain on declining freshwater sources, with some developers suggesting that 30 times less water is required per cubic metre to produce mass timber when compared with reinforced concrete [27]. One recent study in China used a full life cycle assessment approach to compare the emissions profile of an existing seven-storey concrete building with its hypothetical mass timber equivalent. The study concluded that buildings could reduce energy consumption by over 30% and $CO_2$ emissions by over 40% by using mass timber rather than concrete and steel [28]. A separate study in China used modelling to determine that mid-rise residential buildings constructed using mass timber rather than concrete

could reduce energy demands by around 30% and emissions by around 25% during the operation phase alone [29]. These studies indicate that MTC has the ability to improve the energy efficiency of buildings and provides an opportunity for the construction industry to meaningfully engage with the objective of achieving net-zero carbon emissions.

There have been a number of buildings constructed with mass timber over the last two decades that highlight the environmental benefits of its use [7]. The most notable example in Australia is the Forté building, which is a 10-storey mass timber apartment building located in the Victoria Harbour District in Melbourne [27]. Forté was a landmark achievement in MTC, being the first mass timber building in Australia and the tallest wood apartment building in the world when in was completed in 2013. The building was constructed using 485 tonnes of mass timber imported from Europe [27]. It is estimated that the project saved 1451 tonnes of $CO_2$ emissions and has a 22% lower overall carbon footprint when compared with an equivalent reinforced concrete structure [27,30]. Although the carbon footprint of the building is relatively small, emissions could have been reduced further by using locally sourced and manufactured timber as opposed to timber imported from Europe [30].

Another monumental achievement in MTC is the Brock Commons Tallwood House located at the University of British Columbia in Vancouver, Canada [31]. This innovative 18-storey (53 m) high-rise student accommodation facility comprises a hybrid structure, with a concrete ground floor supporting 17 storeys of mass timber floors and columns, as well as two 18-storey concrete elevator cores [31]. It is estimated that the timber used in the building stores 1753 tonnes of $CO_2$ and avoids the production of 679 tonnes of greenhouse gas (GHG) emissions [32]. The savings in GHG emissions realised by using mass timber on this project are equivalent to taking 510 cars off the road for one year. Moreover, the emissions profile of the building could have been reduced further by using an alternative structural system. A study by Connolly et al. (2018) demonstrated that it would have been structurally feasible to construct the two elevator cores in the Brock Commons building using mass timber rather than concrete, which would have further reduced the environmental footprint of the building [33].

The T3 office building in Minneapolis, often referred to as T3 Minneapolis, was a milestone for MTC in the United States [34]. The seven-storey commercial building was the first modern mid-rise timber building constructed in the United States for over a century and, at the time of its completion in 2016, was the largest mass timber building in North America. The design of the building used a structural system that incorporated exposed Glulam beams and columns and NLT floor and roof panels to create an aesthetically pleasing and healthier indoor environment [34]. An NLT system was chosen for this project because it had structural advantages and faster procurement times, as well as lower costs, relative to other mass timber products. In total, the T3 structure is comprised of around 3600 m$^3$ of wood, which sequesters 3200 tonnes of $CO_2$ for the life of the building. In combination, these examples demonstrate that it is possible to construct multi-storey buildings using mass timber and that doing so results in clear environmental benefits.

*3.2. Sustainable Forestry Management*

The potential widespread increase in the use of mass timber in the construction industry does raise concerns about potential deforestation and the depletion of global forest resources, particularly primary forests [35]. This is an important concern given that the increased demand for engineered wood could lead to a loss in global forested areas if sustainable practices are not implemented. There is a growing body of research around sustainable forestry harvesting, which is a process that requires careful long-term planning to ensure that forests maximise their social, environmental and economic benefits into the future [36]. Although the removal of forest carbon stocks has historically been viewed as having negative effects on the environment, recent studies have shown that regularly harvested wood can help reduce carbon emissions [35,36]. The carbon sequestered by forests occurs through the accumulation of biomass within trees. This carbon is stored over

the lifetimes of the trees and is only released when a tree dies and decays [36]. Since mass timber preserves the wood, it has the potential to ensure that the carbon stored by trees may be kept out of the environment for longer than it would if the normal life-cycle of that tree was considered.

Although the uptake of mass timber by the building industry may be viewed as a gateway to diminishing forested areas around the world, significant volumes of timber can be harvested without depleting or degrading forest resources if sustainable harvesting practices are undertaken [35]. The improvement of forestry resources has been demonstrated in regions such as Asia, North America, and Europe, where 75,000 Mm$^3$ of roundwood logs have been extracted since 1990, while forest covers have increased by around 1 Mha/yr [35]. For example, despite the fact that 90% of global CLT production—estimated at 700,000 m$^3$/yr—is located in Europe, the region's forest area increased by 90,000 km$^2$ between 1990 and 2015 under sustainable forestry initiatives [15,35].

Consequently, increased demand for forest products under sustainable management would incentivize new tree planting and investment in forest management that would expand forest carbon sinks by encouraging forest growth and regeneration [37]. That is, as long as forests are managed sustainably, most of the projected depletion in aggregate forest stock due to mass timber-induced increases in removals will be replaced by biological forest growth occurring over time [38].

Sustainable forestry practices include management approaches that ensure current needs for timber are met while not compromising the ability of future populations to make use of the forest resources (wood or otherwise). Given the expected rise in population growth and international pressures to convert forests into agricultural land for farming and commercial use, deforestation is likely to increase in coming years if efforts are not made to conserve natural forest areas and increase rates of afforestation [35]. The IPCC has recognised this problem and stated that a sustainable forest management strategy that aims to increase forest carbon stocks and produce an annual yield of timber will generate the largest emissions mitigation benefit [1].

### 3.3. End-of-Life Scenarios

The adaptation of mass timber panels for other uses at the end of a building's design life is a crucial aspect of the regenerative approach to sustainability [39]. Given that the decomposition or burning of wood releases stored $CO_2$ back into the atmosphere, repurposing mass timber products at end-of-life will be essential for maximising the environmental benefits of mass timber [12]. As such, the three main options for mass timber once a building has been disassembled are re-use, conversion to biomass energy, or landfill [35]. Re-use is the preferred option for mass timber, as it can be used for the same purpose as before or transformed into lower-grade timber for non-structural uses such as facades. The re-use option would extend the life of the timber and maximise carbon sequestration, while reducing the need for new wood and lowering production emissions. If recycling is not possible, the timber can be used to produce biomass energy through direct combustion that does not maximise emissions reduction but does enable a fossil fuel offset. Finally, if both of the aforementioned options cannot be achieved, mass timber can be sent to landfill, though this is the least desirable option as it releases stored carbon and does not recover energy from the wood products [35]. Overall, when conducting a life-cycle assessment for the environmental benefits of mass timber, design for end-of-life scenarios should encourage material recovery and recycling in order to establish mass timber as a truly sustainable material.

### 4. Seismic Performance

Ensuring that buildings can withstand seismic events is one of the most important aspects of building design, especially in earthquake prone regions. Buildings that are not adequately designed to endure earthquakes are susceptible to severe damage or failure that can lead to injury or death for occupants. For example, a catastrophic earthquake hit central

Mexico in 2017 destroying more than 44 buildings and killing around 230 people [40]. As is the case with almost all other seismic events, this devastation was not caused by the earthquake itself, but by the collapse of buildings. Thus, designing buildings to be earthquake-resistant is crucial for reducing damage and increasing safety. Given that mass timber is a relatively new building material, ensuring that architects and engineers understand its structural characteristics and seismic performance is vital for accelerating its widespread adoption. Although the seismic performance of conventional tall buildings has been understood for some time, research into how mass timber buildings perform under seismic actions has only begun within the last two decades [41]. Consequently, research into how mass timber structures respond during earthquakes is somewhat limited, although there are a handful of studies that have demonstrated the viability of mass timber buildings under seismic conditions.

An early research project by Ceccotti (2008) was instrumental in providing evidence that mass timber structures can withstand earthquake events. The study sought to analyse the seismic performance of a three-storey Cross Laminated Timber (CLT) building by undertaking a number of shaking table tests in a specialised facility in Japan [42]. The shaking table tests exposed the test building to a series of earthquake excitations to establish the capability of the structure to dissipate energy efficiently and ensure that the structure can survive extreme earthquakes. The results of the study demonstrated that the CLT test building performed very well during the shaking table tests and did not incur any major damage. Even though the building was repeatedly subjected to 15 destructive earthquakes without repair, including an earthquake producing a near-collapse state, at the end of the entire series there was no permanent deformation and the building remained upright. This research study was especially important for validating the seismic behaviour of mass timber structures.

Following on from the previous study, Ceccotti et al. (2013) published the results of a similar seismic study on a test mass timber building [43]. In this case, a full-scale seven-storey CLT building, designed and constructed according to the European seismic standard 'Eurocode 8', was subjected to a simulated earthquake loading on a 3D shaking table. After enduring a series of major earthquakes, the test building showed no residual displacement and no critical damage. Some level of failure was observed at the hold-down metal fasteners that were used to connect the mass timber panels, with the fasteners being loosened or nails being removed as a result of the seismic excitations. However, these connections can be easily repaired after an earthquake event, whereas conventional reinforced concrete buildings are more likely to undergo critical failure due to their brittle nature.

To gain a greater insight into how mass timber buildings respond during seismic events Shahnewaz et al. (2017) undertook a study in Vancouver, Canada, which investigated the seismic behaviour of a hypothetical six-storey CLT platform building using Incremental Dynamic Analysis [44]. Using this technique, the researchers were able to examine the structural response of the case study building under simulated earthquake excitations. The aim of the study was to verify whether or not the mass timber building could withstand a Maximum Credible Earthquake (MCE) (also commonly referred to as 'Maximum Considered Earthquake'), which is an extremely high intensity seismic event with high level ground motions that are expected to occur roughly once every 2500 years. The results of the analysis indicated that the CLT building would be unlikely to incur any damage during an MCE, and the probability of collapse was found to be less than 0.1%. Hence, the study concluded that mass timber buildings are likely to have a sufficient factor of safety against collapse during major seismic events [44].

More recently, a study was conducted in the USA to explore the advantages and disadvantages of MTC for high-rise buildings in high seismic regions [45]. The study used computational and numerical analyses to compare the existing 20-storey reinforced concrete Museum Tower Apartment building in Los Angeles with a theoretical mass timber equivalent. The mass timber building incorporated glulam columns and beamless composite concrete CLT floor slabs, and was designed to have the same footprint as the

existing reinforced concrete building. The study found that the mass timber building had roughly half the mass and half the stiffness of the reinforced concrete building, which is desirable for seismic design.

These research studies suggest that well-designed mass timber structures can not only exhibit satisfactory performance under seismic conditions but can outperform traditional concrete structures. The superior performance is possible because mass timber buildings are extremely lightweight compared to concrete buildings, thereby minimising inertial forces generated during an earthquake and reducing the risk of failure [46]. Additionally, the high in-plane stiffness of mass timber panels such as CLT allows the structures to resist lateral distortion and ductile connections can yield without compromising the structural integrity of the building [46]. Although research to date has established the viability of MTC under seismic conditions, further research is required to investigate the most appropriate seismic design for tall mass timber buildings greater than 20-storeys.

## 5. Wind Performance

Although it can be beneficial from a seismic performance perspective, the relative flexibility (in comparison to other building materials) of wood could make buildings that are constructed from wood susceptible to wind-driven oscillations that are discomforting to occupiers [47]. Despite the potential importance of this performance indicator however, to date relatively little work has been undertaken to investigate this topic.

The most extensive investigations into the topic of the response to wind of wood building has been undertaken by researchers in Canada, who used a combination of modeling and wind tunnel testing to investigate various components of the relationship between tall mass timber buildings and wind load [48–50]. Collectively, this work indicates that mass timber buildings can be constructed to meet building code requirements for wind drift, although vulnerability to wind impact is strongly dependent on building height. This research provides specific recommendations for the construction, siting and use of 10-, 20-, 30- and 40-story mass timber buildings based on wind performance [50]. This research also involved a preliminary exploration of the structural performance of wood buildings under tornadic conditions [51], revealing that building orientation and shape play important roles in determining wind impact, and that 10-story mass wood buildings could suffer extensive damage if exposed to an EF3 or stronger tornado. In summary, wind performance is an important criterion to consider when designing and constructing tall mass timber buildings, but more work is required to clarify the precise relationships between dynamic wind conditions and mass timber building performance. However, early results suggest it is possible to use these materials to meet building code requirements.

## 6. Fire Performance

Fire safety is a critical aspect of construction across all building materials [52]. One of the major factors limiting the implementation of tall timber structures is the negative perception of wood with respect to fire safety [53,54]. Fire concerns surrounding timber remain one of the key reasons that severe height limitations and building code restrictions exist for wooden buildings around the world [54,55]. However, these fire concerns largely stem from a lack of understanding about the fundamental difference between light-wood frame construction and Mass Timber Construction [39]. In light-wood frame construction, the structural elements of the building are made up of small timber members with significant air voids between them that can allow fire to spread and engulf the building, leading to structural collapse [39]. However, with MTC, solid wood panels with large section sizes are designed to minimise air voids and resist fire damage [39]. Although there are still some concerns amongst industry professionals in relation to the fire performance of mass timber, there has been a significant amount of research in this area and different approaches to ensure a fire safe design are well established. Appropriate structural design methods, along with studies containing experimental results from multiple fire tests, are reviewed and presented in this section.

*6.1. Charring Method*

Although it is considered a combustible material, mass timber burns in a slow and predictable manner [56,57]. It is well understood that when exposed to flame, the outer layer of wood ignites and burns, removing hydrogen and oxygen from the surface and forming a charred layer comprised predominantly of carbon [56,57]. Beneath the charred layer, a section known as the 'pyrolysis zone' or 'zero-strength layer' is formed, where the increase in temperature leads to decomposition of the wood in this layer. The residual cross-section is left largely unaffected by the fire, given that the charred layer acts as an insulator for the remaining wood and provides a thermal barrier between the exposed surface and the inner core [15,57].

Many experiments on the fire behaviour of mass timber members have verified that both the charred layer and the zero-strength layer lose their structural integrity during and after a fire, whereas the inner core retains its full load-carrying capacity [52]. Hence, exposed structural members in mass timber buildings can be designed to include a sacrificial layer that would protect the inner core of the member in the case of a fire and prevent the structure from collapsing. The dimensions of each member can be designed using a predictable charring rate so that the required load-carrying capacity is not compromised in the case of a fire [39]. The charring rate for heavy timber has been tested in many studies over the last few decades [54,58,59]. The charring rate for solid wood panels under standard fire exposures is given as 0.65 mm/min for both CLT and glulam members, which has been recognised by European and North American building codes for years [58]. For unprotected timber surfaces, the measured charring rate is assumed to be constant with time and can therefore be used in design calculations to ensure an adequate level of fire resistance for structures with exposed mass timber [58]. This approach is known as the charring method, or reduced cross-section method, and has been used in a number of real mass timber buildings to safely include exposed wood.

Despite the established charring rate of solid wood, a number of associated factors must be taken into consideration when designing exposed mass timber finishes. An important earlier study testing the behaviour of CLT panels in fire showed that specimens manufactured with temperature sensitive adhesives such as polyurethane (PUR) can result in char fall-off [58]. Char fall-off (also known as 'delamination') refers to a process within a CLT panel where the char depth increases as the wood burns and under the action of gravity the charred wood at the glue line can no longer remain adhered to the panel [60]. Given the influence of gravity, this effect is more likely to occur for floor and ceiling panels than for wall panels [61]. Once char fall-off has taken place, the charred layer is no longer able to protect the underlying timber from heat transfer, resulting in an increased charring rate and prolonging the fire [58]. The study found that the use of less temperature sensitive adhesives such as melamine urea formaldehyde (MUF) resulted in no char-fall off. A recent study confirmed that CLT floor assemblies that utilise PUR adhesive bonds tend to soften and result in delamination whereas panels bonded with MUF remain intact during extended fire exposure [54]. Another vital consideration when using the charring method is that the number and thickness of the layers that make up the mass timber panel significantly influences the fire resistance of the member. Multiple studies have verified that CLT panels with five or seven layers of greater thickness exhibit better fire performance than those with fewer and thinner layers, especially for polyurethane adhesives [58,62]. Hence, to achieve an adequate level of fire safety using the charring method, it is essential that designers consider using large mass timber members with thick layers [54,58], although being manufactured with more fire-retardant adhesives such as MUF may also need to be weighed against off-gassing health issues upon initial construction [18].

Although the charring behaviour of mass timber under standard fire exposures is well documented, the structural response of load-bearing members under non-standard heating is not as well understood. Current practice using the charring method assumes that the thickness of the zero-strength layer is a constant 7 mm beneath the charred layer [63]. However, this approach has been criticised for being inaccurate and unrealistic for solid timber, and recent studies have confirmed that a constant 7 mm zero-strength layer is not applicable for non-standard fire exposures [59,63]. Thus, based on limited existing studies regarding the formation of the zero-strength layer, additional research is required to develop a more accurate method to calculate the thermo-mechanical response of mass timber members, especially under non-standard fire exposures.

*6.2. Encapsulation Method*

An alternative method for ensuring fire safety in mass timber buildings is to encapsulate members with fire-rated materials such as gypsum plasterboard [39]. Using this technique, one or two layers of fire-rated gypsum boards are installed directly onto the mass timber panels to protect the structural elements of the building in the case of a fire [39]. The purpose of this approach is to protect the underlying members and prevent the structural mass timber from contributing to the fire load [55]. The encapsulation method is a more conservative fire design approach than the charring method; however, it is deemed an acceptable solution in most national building codes and is recognised as being able to provide an equivalent level of safety to non-combustible construction such as steel or concrete [39,53].

The encapsulation method can be applied to mass timber design in two ways. Complete encapsulation would require that all structural mass timber is protected by gypsum boards of a sufficient thickness (two layers) so that the underlying wood is not adversely affected by a fire. The alternative approach is limited encapsulation, which entails providing one layer of thin gypsum plasterboard that protects the structural timber until well into the burning phase but may not prevent the mass timber from charring. Limited encapsulation is a more economical solution; however, the level of encapsulation should be chosen based on the requirements of the relevant building standards [55].

*6.3. Additional Experimental Results*

There have been many studies over the last two decades that have investigated the fire behaviour of mass timber. These studies provide results from a range of different experiments on single mass timber assemblies and full-scale compartment fire tests with and without encapsulation. The purpose of these standardised fire tests is to ensure that the building material can withstand a fire and retain its structural integrity for a specified duration [39]. The requirement for fire duration varies depending on the national building standards but is typically between one and two hours, which gives occupants enough time to safely exit the building and allows firefighting services time to suppress the fire and prevent further damage to the property [39]. A brief overview of the results from a series of experimental studies testing the fire behaviour of mass timber assemblies in different configurations as well as full-scale compartment fire tests is outlined in Table 2.

**Table 2.** Summary of experiments and tests analysing the fire performance of mass timber.

| Authors | Investigated Parameters | Results |
|---|---|---|
| Frangi et al. (2008) [56] | 1-hr compartment fire test on full-scale three-storey CLT building with encapsulation | Structure passed the 1-h fire test Fire spread was limited to one room and no elevated temperature or smoke was detected in the room above the fire compartment |
| Frangi et al. (2009) [58] | Fire tests on CLT panels with various thicknesses and adhesives | Char fall-off was observed for panels bonded with a temperature sensitive adhesive Panels bonded with a less temperature sensitive adhesive exhibited better fire performance |
| Osborne et al. (2012) [64] | Fire tests on eight different CLT wall and floor panels with and without encapsulation. All members were subjected to imposed loads | All but one test passed the 1-h fire rating A 7-ply floor assembly was able to withstand the fire for close to 3-h before reaching structural failure |
| Zhao Peng and Pei Feng (2012) [65] | Fire tests on eight different glulam beams | All tests achieved a fire rating over 90 min and exhibited good fire behaviour Large section members performed better than smaller members Intumescent coatings can be used to improve the fire resistance of the member |
| Aguanno (2013) [66] | Fire tests on eight different CLT floor assemblies with and without encapsulation | All but one test passed the 1-h fire rating Although the encapsulated panels performed better overall, the 5-ply exposed panels achieved a fire rating of over 90 min |
| Klippel et al. (2014) [61] | Fire tests on ten different CLT wall and floor panels without encapsulation and subjected to imposed loads | Measured charring rate was found to be slightly higher than 0.65 mm/min for almost all tests; however, this is attributed to the falling off of charred layers due to the use of a temperature sensitive adhesive |
| Hasburgh et al. (2016) [53] | Fire tests on 23 CLT assemblies with encapsulation using different fire-rated materials | All investigated materials greatly delayed the onset of charring to the mass timber and resulted in adequate fire performance |
| Muszynski et al. (2019) [54] | Fire tests on three different CLT floor assemblies without encapsulation and subjected to imposed loads | All three unprotected floor assemblies passed the 2-hr fire rating following ASTM E119 standard procedure |

### 6.4. Fire Performance Summary

Fire safety is one of the most important design considerations for any building. Given that it is a combustible material, wood has historically been perceived as having an inadequate level of fire safety and, as such, building code restrictions have prevented its use in tall buildings [55]. However, new provisions for mass timber are slowly being introduced into building codes in many countries, given the extensive research data that supports its adequate fire performance [67]. Many studies have verified the charring effect of heavy timber elements in a fire, which if designed appropriately allows exposed mass timber to be incorporated into a building. This results in a fire safe design that showcases the natural beauty of wood and increases the health benefits of the indoor environment for occupants. To achieve a higher level of fire safety, mass timber panels can be encapsulated in fire-rated materials such as gypsum plasterboard where necessary, which protects the underlying members in the case of a fire and largely prevents any damage to the structure.

Note that the fire design methods discussed here are purely passive protection measures and do not take into account the required installation of active protection measures, such as automatic sprinklers and smoke detection systems. In most cases, automatic sprinkler systems are the most effective method for improving the fire safety of buildings given that they are able to extinguish the fire well before any damage occurs [55]. Further, the risk of fire damage to tall wood buildings will be reduced assuming that fire department resources are promptly dispatched. Hence, using a combination of active and passive protection measures, mass timber buildings can easily achieve an appropriate level of fire safety.

### 7. Health Effects and Biophilia Benefits

To ensure the health benefits of mass timber, recommendations for moisture mitigation in products such as CLT should be followed, especially in terms of good moisture management during construction, an objective that is facilitated by the reduced construction time of such prefabricated materials [68]. Similarly, properly installed CLT products have been found to have a negligible impact on indoor air quality in terms of volatile organic compounds, although some wood species still need testing [18].

A crucial aspect of mass timber that is not widely discussed in the literature is the positive affect that it can have on people [69]. Natural elements such as plants and trees in outdoor settings provide a range of social and health benefits within towns and cities. Numerous studies have demonstrated that natural environments tend to promote greater interaction between community members, encourage physical activity, lower crime rates, and reduce stress [70–72]. Even something as simple as having a view to nature can promote positive emotions and reduce negative feelings such as depression and anxiety [73]. Similarly, there is a growing body of research indicating that natural elements in indoor environments can improve the mental and physical health of occupants [69,74].

The idea that humans have an affinity for natural environments and an affection for plants and other living things is referred to as 'biophilia' [75]. This innate desire to be surrounded by nature is likely a result of the evolutionary history of the human species and the importance of fauna for our survival [76–78]. In other words, humans have evolved to live in a natural environment. However, people in industrialised countries currently spend around 90% of their time indoors where there is a substantial lack of natural stimuli [79]. As such, the negative effects that sterile indoor environments have on building occupants is significant and there is growing awareness amongst designers and industry professionals of the importance of healthy and sustainable buildings [69].

Given the recent shift towards biophilic design principles, new buildings are commonly featuring elements such as views to nature, natural sunlight, indoor plants and water features, all of which have proven physiological and psychological benefits [80]. A number of studies have also shown that workplaces that include natural elements such as indoor plants tend to result in fewer health complaints from employees and a decrease in reported sick days [75]. Hence, it is well substantiated that interacting with nature has positive effects on health and well-being.

Similar to other natural elements, wood represents a connection to trees and nature that offers a number of health benefits for occupants [81]. It has been observed that people tend to have a positive attitude towards wood, perceiving it as a natural, warm, and healthy material [69,82]. However, only in recent years has research demonstrated the quantifiable benefits of interior wood use for building occupants [80]. For example, a study was conducted by Fell (2010) to test the stress effects of wood within the context of an office setting [76]. Wood and non-wood offices were presented to 119 subjects and stress responses were measured by testing pulse rate and skin conductivity. The results of the experiment indicated that subjects in the wood room were less stressed than subjects in the non-wood room. Hence, the study provides evidence that wood produces stress-reducing effects similar to the benefits of exposure to nature.

In a similar study, stress levels of Austrian high school students working in wooden and non-wooden classrooms were measured over the course of a school year [83]. The control condition was a typical classroom containing plasterboard walls, a linoleum floor, and chipwood cupboards. The experimental condition was a classroom made almost entirely from solid wood (i.e., walls, floor, ceiling, and cupboards). Over the course of the school year, researchers found that student heart rates and stress levels decreased in the wooden classroom. Moreover, the wooden classroom was observed to be positively associated with increased concentration and healing, as well as reduced strain, providing further evidence that wooden indoor environments lead to positive health benefits [83].

In a more recent study undertaken by Zhang et al. (2017), an experiment was conducted to assess the physiological effects of wooden and non-wooden indoor environments on participants [84]. To simulate the contrasting indoor environments, four identically sized rooms with different interior walls were prepared. All participants were continuously monitored to test for physiological indicators. The experiment found that participants consistently exhibited lower systolic blood pressure and heart rate in the wooden rooms compared with the non-wooden room. Moreover, tension and fatigue were significantly reduced in the wooden rooms when participants completed their tasks. The results of the study indicate that wooden indoor environments have a positive effect on the autonomic nervous system, respiratory system, and visual system and play an active role in reducing stress and creating a visual relaxation effect [84].

## 8. Cost Analysis

One of the key reasons for the slow uptake of Mass Timber Construction (MTC) around the world is the fear that this new methodology will result in higher project costs than traditional reinforced concrete construction [85]. In a number of recent studies surveying industry professionals on their opinions of mass timber and the perceived barriers to widespread adoption, higher cost was among the most frequently cited concerns [16,86]. Given that the building industry is highly risk averse, the implementation of new technologies or construction processes is unlikely to occur unless a quantifiable cost saving can be achieved [87]. Innovations that deliver projects on-time and under-budget are the main drivers of competitive advantage in the construction sector [87]. Hence, it is critical that industry stakeholders understand the many financial advantages of using mass timber.

The expected costs associated with the use of mass timber are not well understood as it is a relatively new building material and there are a limited number of real-world examples that can demonstrate construction costs and program benefits [85]. However, there have been many case studies in recent years that have presented business cases for mass timber and compared project costs to traditional construction projects utilising steel and concrete. This section reviews these studies and provides an outline of the economic advantages of MTC projects across various influencing factors such as materials, labour, scheduling and economic growth opportunities.

### 8.1. Material Costs

The material costs for a building project make up a significant portion of the overall construction costs [88]. As such, it is important that mass timber is priced similarly to steel and concrete if it is to be considered a viable alternative to developers in a competitive construction market. However, despite its importance it can be difficult to directly compare material costs due to issues such as material availability and market supply and demand [88]. Nonetheless, several recent studies have presented a comparison of the material costs related to the structural frame of mass timber and conventional concrete buildings with mixed results.

A study assessing the nine-storey Murray Grove Cross Laminated Timber (CLT) building in the U.K. found that using a mass timber structural system resulted in a 30% increase in material costs compared with reinforced concrete [89]. A similar study by Fanella (2018) compared costs of a hypothetical 10-storey CLT building in the U.S. to a cast-in-place

concrete solution [90]. The results of the study found the cost of the CLT option to be 16–29% higher than the concrete system. However, other studies have shown that cost savings can be achieved using MTC compared with traditional methods. A detailed cost analysis of a hypothetical seven-storey office building in Sydney, Australia demonstrated that a mass timber design would save 13.6% when compared with a concrete system [91]. All structural components of this building were found to be more cost effective using mass timber except for the Glulam columns. Green (2017) came to a similar conclusion, determining that structural walls for a mass timber building would be 26% cheaper to implement, whereas upper floors including columns, beams and finished materials would cost 43% more than a concrete design [39]. The source of the materials must also be taken into consideration, as one study in the U.S. showed that cost savings for mass timber projects could be doubled if the materials are produced locally as opposed to being imported from Europe [92]. Additionally, given the lightweight nature of wood, mass timber buildings are typically 40–50% lighter than equivalent concrete buildings [15]. Consequently, mass timber buildings require less foundation concrete to create smaller and lighter foundations, which significantly reduces earthworks and foundation costs [93].

It is not definitively clear from the existing research whether or not mass timber is more cost effective than cast-in-place concrete when isolating material costs. However, even if it is assumed that material costs are currently higher for mass timber than for steel or concrete, it is expected that these costs will reduce as the design and development of mass timber buildings improves and market supply chains mature [15]. Furthermore, considering only material costs ignores the costs savings that can be realised in other areas of construction projects [91].

*8.2. Labour Costs*

An element of MTC that can provide significant cost savings is the reduction in on-site labour [94]. The innovative manufacturing process for mass timber allows structural components to be prefabricated off-site and assembled on-site by a small team of labourers, directly reducing costs of on-site trades [95]. For example, the Forté building that was completed in Melbourne in 2013 only required five skilled labourers and one supervisor on site during the construction process [27]. Similarly, the eight-storey Bridport House apartment building in London constructed in 2011 only needed a team of four skilled labourers and one supervisor [94]. The substantial reduction in on-site labour and subsequent labour costs will obviously lower the overall costs of mass timber projects.

*8.3. Scheduling*

One of the greatest overall advantages of MTC is the speed with which mass timber buildings can be erected [94]. As mentioned previously, the prefabrication of mass timber panels allows the structural system of the building to be delivered to site where it can be installed very quickly by a small crew of workers [96]. Not only does this speed up the construction phase, but it also allows for the structural components of the building to be constructed concurrently with the foundations and footings [94]. This reduces the lag time that exists for conventional building projects where ground improvements must be done before the structural frame can be built. In addition, there is a considerable lag time that exists for traditional building projects given that concrete must be cured for at least 28 days before its final strength is reached and construction of the next floor can begin [96]. This lag time is largely eliminated on MTC projects, with one study suggesting that a mass timber building can be erected at a rate of 3–4 days per storey, as opposed to 28 days per storey for a typical reinforced concrete construction [97]. Hence, if design, manufacture and construction on a mass timber project are coordinated appropriately, time on-site is drastically reduced, which has a number of added benefits including increased safety for workers, less disruption to the surrounding community, and reduced material waste [96].

Many case studies have demonstrated the superior building speed of MTC compared to traditional methods. A study on mid-rise residential buildings in Melbourne found that a time saving of close to 50% can be achieved when designing a Laminated Veneer Lumber (LVL) superstructure as opposed to a full reinforced concrete structure [85]. Another study looked at the differences in costs and construction timelines between an existing arts centre building in Napa, California, and its hypothetical mass timber equivalent. The study found that when cast-in-place concrete and structural steel elements were replaced by CLT and glulam members, the construction timeline was reduced by around 61% [88]. A study by Smith et al. (2018) was important for comparing the schedule performance for a range of existing mass timber buildings to their traditional counterpart [94]. Their study identified that MTC reduced construction schedules by an average of 20% across the seven real case studies, with an average duration of 12.7 months for mass timber projects compared to 15.4 months for typical concrete construction. The study also identified that the Forté apartment building in Melbourne was completed 3 months faster than could have been achieved using traditional methods, whilst the Bridport House mass timber building in London reduced the building schedule by 8 weeks [94]. Although the improvement in construction efficiencies is evident from looking at these case studies, the research tends to suggest that knowledge acquired from accumulated experience working with mass timber will streamline costs and increase productivity even further [94].

*8.4. Economic Growth Opportunities*

An important aspect of MTC that must be considered by government officials and policymakers in order to accelerate the large-scale adoption of mass timber is its potential to provide economic benefits into the future [93]. Given the reduced need for skilled labourers on mass timber projects, state level policymakers may be hesitant to endorse MTC as it would result in employment losses for tradespersons and pushback from construction unions. However, mass timber has the potential to reshape the building industry and provide opportunities to stimulate local job growth in areas facing forest products manufacturing decline [93]. One study in the U.S. analysed the regional employment opportunities and economic growth that mass timber could create in the state of Oregon. The study found that if mass timber gained 5% of the region's construction market share, increased product demand would generate around 2000 manufacturing jobs, and if mass timber constituted 15% of the market share, this would increase to 6100 direct jobs [98]. Another study assessing the regional economic impacts of a 12-storey high-rise MTC building in Portland, Oregon found that by using a mass timber structural frame, as opposed to a functionally equivalent concrete frame, the building was able to create an additional $2.39–$4.97 million in economic activity and produce greater earnings for households of all income levels [93]. It was also found that economic benefits are maximised when mass timber panels are manufactured locally rather than being imported, creating a more insular supply chain and reducing economic leakage [93]. Thus, although there may be concerns that MTC will lead to job displacement in the construction sector, growth in the forestry industry and a renewed need for harvesting and manufacturing jobs due to the rise of mass timber will provide vast employment opportunities for industrialised regions and will boost local economic output in a manner that promotes environmental sustainability.

*8.5. Summary*

The findings presented in this section clearly outline the economic and financial advantages of MTC and its potential to become even more cost competitive with wider adoption in the building industry. In relation to the factors influencing the cost analysis, it was found that in many cases material costs for mass timber were slightly higher when compared directly with reinforced concrete. However, these material costs are offset by other factors such as reduction in labour costs, foundation costs, and project timelines. When comparing overall project costs, many studies have cited an average 4% cost saving on mass timber projects compared with traditional construction [94,99]. This cost sav-

ing indicates that mass timber can be a cost-effective alternative, and research suggests that growing acceptance and implementation of this emerging technology will further reduce costs [93]. Furthermore, MTC provides an exciting opportunity to revitalise sustainable forestry harvesting and improve economic outcomes whilst simultaneously working towards delivering a net-zero built environment.

## 9. Discussion

This paper reviewed the existing literature to investigate the performance of Mass Timber Construction (MTC) relative to conventional reinforced concrete and steel systems. The main goal of this paper was to investigate the viability of wood as an alternative to steel and concrete in construction. The review indicates that mass timber meets all engineering requirements for structural applications within buildings, and even outperforms traditional construction materials in several ways. In relation to sustainability, mass timber significantly reduces life-cycle emissions, air pollution, energy usage and water usage compared to concrete and steel. Further, given the carbon sequestration properties of preserved wood, mass timber is likely to be the only building material that could substantially reduce greenhouse gas (GHG) emissions and play an important role in slowing down climate change.

This review investigates the potential for using wood as a building product, considering environmental, seismic, fire, economic and health performance; but how does this product compare with more traditional building materials? Table 3 provides a summary of the performance of MTC relative to reinforced concrete construction across all of the key performance criteria outlined and reviewed in this study. Results from several experimental studies demonstrate that the extremely high strength-to-weight ratio of mass timber and high dimensional stiffness allow tall timber structures to withstand earthquake events and, despite the fact that wood is a combustible material, mass timber panels can be designed to meet traditional building code requirements for fire design. With respect to the effects of mass timber on the health of building occupants, the research indicates that wooden indoor environments are commonly perceived as natural, warm, and healthy and that interior wood use significantly reduces stress and blood pressure while improving cognition, productivity and emotional condition. Finally, an analysis of costs from a number of existing mass timber projects found an average cost saving of 4% when compared with conventional construction methods. In particular, the prefabricated nature of the timber construction process and the ease of assembly on site result in lower costs in relation to foundations and earthworks, on-site labour, heavy machinery, and project timelines. Furthermore, costs of mass timber projects are expected to decrease in the future as product uptake is increased.

Table 3 indicates that MTC has many advantages over concrete and steel and that some of these benefits are substantial. These advantages suggest that MTC should be considered a viable alternative to existing conventional construction materials and, arguably, the areas in which it excels, such as reducing carbon emissions and promoting mental and physical health, serve to make it an extremely desirable product.

**Table 3.** Performance of Mass Timber Construction (MTC) relative to conventional construction.

| Performance Criteria | Performance Rating |
|---|---|
| *Environmental* | |
| Carbon Emissions | Far Better |
| Energy Usage | Far Better |
| Water Usage | Far Better |
| *Seismic* | |
| Seismic Behaviour | Better |
| *Wind* | |
| Wind Performance | Undetermined |
| *Fire* | |
| Charring Method | Similar |
| Encapsulation Method | Better |
| *Health* | |
| Mental Health | Far Better |
| Physical Health | Far Better |
| *Costs* | |
| Material Costs | Similar |
| Foundation and earthworks | Far Better |
| Labour Costs | Far Better |
| Speed of Construction | Far Better |
| Economic Growth Potential | Far Better |

## 10. Conclusions

Although this paper reviews research into many aspects of mass timber use as a building material, and demonstrates the feasibility of MTC, further research is required to increase its uptake and change international and local building codes to include specific requirements for timber structures. Important safety design needs, such as structural characteristics, seismic behaviour, wind resistance and fire resistance, will benefit from research that offers solutions for every possible design combination. Given that mass timber is still in its infancy in the construction industry, investigations into ways of optimising its use in structural systems, and how it can be combined with steel and concrete to create hybrid structures, will be very valuable. To accelerate the transition to more sustainable practices across the construction industry in general, it is recommended that governments start adopting mass timber for major infrastructure projects and encourage its use where possible. Similarly, a key component of expanding the utilisation of mass timber will be developers and contractors capitalising on the potential of wood construction and proving that it provides a competitive advantage to conventional building materials.

**Author Contributions:** Conceptualization, J.A. and S.R.; methodology, J.A. and S.R.; formal analysis, J.A. and S.R.; investigation, J.A. and S.R.; writing—original draft preparation, J.A. and S.R.; writing—review and editing, J.A., S.R., M.N. and J.R.; supervision, S.R. All authors have read and agreed to the published version of the manuscript.

**Funding:** This research received no external funding.

**Institutional Review Board Statement:** Not applicable.

**Informed Consent Statement:** Not applicable.

**Conflicts of Interest:** The authors declare no conflict of interest.

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
