# Peer review of "A Review of the Performance and Benefits of Mass Timber as an Alternative to Concrete and Steel for Improving the Sustainability of Structures"

_sustainability, doi:10.3390/su14095570_

Round 1

Reviewer 1 Report

Undoubtedly the manuscript was written interestingly, but its scientific side is fragile.
"Sustainability" is a scientific journal, not a modern construction and building technology review. When reading the article titled "Reducing Carbon Emissions in the Construction Industry - The Benefits of Mass Timber" I expected something more scientific and not just examples of wood used as a building material
The title of the manuscript suggests doing calculations on carbon emission comparisons using wood and other materials as building blocks. The use of wood in construction and its ability to be recycled, which binding the carbon taken by trees from the atmosphere for many years, is a given. The advantages of wood as a building material are also obvious. Writing about obvious things is not a topic for articles in scientific journals. 
The article needs to be rewritten and focus on the environmental effects of using wood construction technologies, including the negative impact on the environment: carbon dioxide emissions resulting from the machinery used and energy consumed to wood processing and to the production of adhesives, and other materials used in this technology. 
The article lacks a discussion chapter. The authors have no reference from the opposing side to the proposed technologies using wood as a building material.

Reviewer 2 Report

The manuscript entitled “Reducing carbon emissions in the construction industry – the benefits of mass timber” deals with a deep analysis of the usage of timber as an alternative of concrete and steel, specially focused on multi-story buildings. The assessment concludes that this kind of material overcomes or is similar to concrete & steel upon environmental, seismic, fire, health and costs comparisons.

The article is very interesting and show in a realistic manner the possibility of using wood in the construction sector of the world. Nevertheless, as the same authors remark, this material is not used yet in an extensive way. In the same sense is my first comment:

  • It is not mentioned any study focused on one characteristic that any material of construction must provide: security, understood as the level of resistance for certain kind of events or situations such as burglars, trespassing, traffic accidents, hurricanes, tornados etc, not only structural reliability for earthquakes. Timber must provide these resistances, especially in urban areas, where generally these incidents and accidents are more common. Perhaps this influences in the hesitation of the people for using timber.
  • It is not clear which the level of production of timber is necessary to fulfill the world demand in construction. I couldn´t find what would be the time of growing and production of wood, depending on the type of tree, climate conditions, water availability etc. In the document, it was pointed out that in the coming decades millions of houses will be required throughout the world. What would be the projected land of production according to the expected demand? Is there any estimation of time of timber production since the seed is planted?
  • One of the manners to compare the environmental impact of different materials of construction is the Life Cycle Assessment (LCA), which assesses 9 categories of environmental impact such as global warm potential, depletion potential of the stratospheric ozone, formation potential of the tropospheric ozone, acidification potential and eutrophication potential, among others, during the stages of production, construction, operation and end-of-life. Why this analysis was not included in the document since it seems an excellent opportunity of being applied?
  • Lines 56 to 58: “In fact, studies have shown that material selection has a greater influence on building environmental performance than building operation” This sentence is not clear, do you mean the indoor environment or the external environment? In either case, this only happens under certain circumstances. In general, operation stage has the most influencing factors for indoor environment and environmental impact.  

Reviewer 3 Report

This manuscript reviewed the key performance criteria for mass timber and subsequently compared with those for typical concrete construction. The manuscript is rich in content, and analyzes the types, performance, design, and economic technology of mass timber. The manuscript is well-written. And there are some comments for further improving the quality of this manuscript, as presented below:

  1. In section 3.5, could you please provide some pictures about Structural Composite Lumber
  2. In table 1, You state that some mass timber products are expensive and some are economical. To whom is this compared? It is best to provide a typical cost
  3. Lines section 4, interesting introduction about mass timber in the construction application, can you provide some pictures to show
  4. Lines 319-324, Using MTC instead of concrete and steel can reduce carbon emissions, and the amount of concrete and steel substituted here should be indicated
  5. Lines 369-384, Interesting point, but to the best of my knowledge, carbon dioxide is not completely stored in trees. It also has photosynthesis, which converts carbon dioxide into oxygen
  6. How do you think about durability of Mass Timber, long-term exposure to humidity or exposure to the sun or alternating between the two. In addition, whether the resulting smell of mass timber, or the smell itself, affects Health Benefits

Round 2

Reviewer 1 Report

Changing the title of the manuscript fundamentally affected my evaluation. The new title is better suited than the previous title to the content of the manuscript. Changes made to the manuscript by the authors improved its scientific value. 

Reviewer 2 Report

Two points that haven't been clarified yet:

Response 1: Thank you for your support about the very interesting nature of mass timber and the need to help advocate for its use. Unfortunately, aspects such as the burglary susceptibility of mass timber that have not been previously studied, hence they cannot be covered in this review. Of the topics mentioned, hurricane and tornado force winds have so far only been explored by one researcher with limited data. That limited data shows that mass timber holds up favourably, but we do not feel the research area is mature enough to include.

These statements must be included within the manuscript. Even the studies are non-existed or limited, all the aspects of construction with a new material should be addressed in order to provide the sufficient information, negative or positive, for the consumers. Therefore, they will be able to make a decision.

Response 3: This issue has been addressed in our previous response to Reviewer 1 Point 1. This paper is a review of the mass timber literature with a view to demonstrating its relative performance against traditional concrete and steel construction. It is not an LCA paper, as this would be a very different sort of paper with a different focus.

I’ve never asked a full LCA assessment. In my opinion, it is important to stablish why this study is only focused on greenhouse gases and not in other environmental impacts of the timber compared with concrete and steel.

Round 3

Reviewer 2 Report

The authors have correctly addressed my last two concerns. The manuscript can be now considered as suitable for publication.